# Reaching elimination of onchocerciasis transmission with long-term vector control and ivermectin treatment in Togo

**Luís-Jorge Amaral**[1,2] ✉, **Rachel N. Bronzan** [3,4], **Anders Seim** [5],
**Marie-Denise Milord**[3], **Koffi Padjoudoum**[6], **Ibrahim Gado Telou**[7], **Sibabe Agoro**[7],
**Michel Datagni**[8], **Piham Gnossike**[9], **Jonathan I. D. Hamley** [1,10],
**Martin Walker** [1,11] & **María-Gloria Basáñez** [1] ✉

The Onchocerciasis Control Programme in West Africa implemented vector control (VC) and ivermectin mass drug administration (MDA) to eliminate blindness, intensifying efforts in Special Intervention Zones (SIZ). Togo aims to eliminate onchocerciasis transmission (EOT) by 2030. We use the EPIONCHO-IBM model to project microfilarial prevalence trends across Togo's five regions by SIZ status, MDA coverage (65%-80%) and VC efficacy (60%-100%). We compare projections with prevalence surveys (400 villages, 1970–2017) stratified by hypoendemic, mesoendemic, hyperendemic, and holoendemic baseline endemicity, and calculate EOT probabilities for 2024, 2027, and 2030. Combined VC and MDA reduced prevalence nationwide. After cessation of VC, prevalence continued to decline in hypo-to-mesoendemic areas under annual MDA, while hyperendemic areas required biannual MDA. In holoendemic areas, prevalence rebounded even with biannual MDA, indicating that alternative strategies are needed. EPIONCHO-IBM reproduces Togo's onchocerciasis trends throughout five decades of intervention and provides a transferable framework to guide policy towards 2030 goals.

Onchocerciasis, caused by the filarial nematode *Onchocerca volvulus* and transmitted by *Simulium* blackflies, remains a public health concern, particularly in sub-Saharan Africa where mean microfilarial prevalence in 2018 exceeded 25% in some countries[1]. The Global Burden of Disease 2021 Study estimated 1.3 (0.8–1.9) million disability-adjusted life-years, and 20 (18–22) million people infected[2]. Ocular, cutaneous and neuro-hormonal sequelae cause substantial morbidity[3,4] and excess mortality[5]. Interventions include vector control (VC) and mass drug administration (MDA) of ivermectin for

prolonged periods, owing to the 10-year (average) adult worm lifespan[6].

The Onchocerciasis Control Programme in West Africa (OCP, 1974–2002) aimed to eliminate onchocercal blindness through weekly aerial larviciding of *Simulium damnosum sensu lato* (s.l.) riverine breeding sites for ≥14 years[7]. In the late 1980s, ivermectin MDA was introduced alongside VC or, in some Western Extension foci, as the sole intervention[8]. In 2012, the goal shifted from morbidity control to elimination of transmission (EOT)[9].

---

[1]MRC Centre for Global Infectious Disease Analysis, Department of Infectious Disease Epidemiology, School of Public Health, Imperial College London, London, UK. [2]Global Health Institute, University of Antwerp, Antwerp, Belgium. [3]Health & Development International, Newburyport, MA, USA. [4]Gates Foundation, Washington, WA, USA. [5]Health & Development International, Fjellstrand, Norway. [6]National Onchocerciasis Control Program, Kara, Togo. [7]Ministère de la Santé et de l'Hygiène Publique, Lomé, Togo. [8]Health and Development International, Lomé, Togo. [9]Neglected Tropical Diseases Coordinator, Ministère de la Santé et de l'Hygiène Publique, Lomé, Togo. [10]Department of Visceral Surgery and Medicine, and Multidisciplinary Center for Infectious Diseases, University of Bern, Bern, Switzerland. [11]Department of Pathobiology and Population Sciences, Royal Veterinary College, Hatfield, UK. ✉e-mail: luis.amaral20@imperial.ac.uk; luisjtmamaral@gmail.com; m.basanez@imperial.ac.uk

Achieving EOT with MDA largely depends on pre-control endemicity (baseline microfilarial prevalence), determined by vector biting rates[10]. The impact of MDA on transmission is influenced by therapeutic coverage (proportion of population receiving treatment) and adherence (proportion of eligibles consistently taking treatment). It has been proposed that a minimal therapeutic coverage of 65% of total population (80% of eligibles) should be reached and sustained for at least 15–17 years to achieve elimination goals[11,12].

The World Health Organization's 2021–2030 Roadmap on neglected tropical diseases (NTDs) aims at verifying onchocerciasis EOT in 12 (31%) endemic countries by 2030[13]. Verification of EOT requires stop-MDA surveys and, if successful, post-treatment surveillance (PTS) for 3–5 years following MDA cessation[11]. Togo, having eliminated four other NTDs[14], aims to achieve onchocerciasis EOT by 2030[15]. The epidemiology of onchocerciasis is heterogeneous across its five regions, from north to south: Savanes, Kara, Centrale, Plateaux and Maritime (Fig. 1, and Supplementary Information: Text 1.1, Table 1 and Fig. 1), each further sub-divided into prefectures. The blackfly-prolific Oti River Basin and its tributaries (Kara, Kéran and Mô) (Fig. 1, and Supplementary Information: Text 1.2), as well as at-risk hard-to-reach villages pose particular challenges to EOT[15,16].

Onchocerciasis control in Togo began with the OCP in the northwest, gradually expanding southward (Supplementary Fig. 2a and Supplementary Table 2). Savanes, Kara and a portion of Centrale were included in the OCP early VC Phases II and III East (starting in 1976–77). The remaining areas of Centrale, Plateaux and Maritime, were covered by the Southern Extension (starting in 1988–89), and

annual ivermectin MDA was introduced. By the end of the OCP in 2002, Kara and parts of Savanes and Centrale had not achieved satisfactory entomo-epidemiological results[17] and were included in the OCP Special Intervention Zones (SIZ, Supplementary Fig. 2b), receiving aerial larviciding until 2007 and intensified, biannual MDA until 2012[18]. After the OCP/SIZ, Togo continued onchocerciasis control through annual or biannual MDA (Supplementary Information: Text 1.2).

Modelling studies and recent reviews suggest that EOT may be attainable in areas with low to moderate baseline endemicity through sustained, high-coverage annual MDA[19,20]. However, highly-endemic foci will likely require alternative treatment strategies (ATS), such as moxidectin MDA[19,21], potentially reinforced by community-directed "slash-and-clear" VC[22].

Following 2015–2017 stop-MDA surveys in Maritime, similar surveys commenced in Savanes in 2023[23]. To support evidence-based decision-making by the National Onchocerciasis Control Programme (NOCP), we modelled, using the individual-based, stochastic EPIONCHO-IBM transmission model[10], the temporal trends of *O. volvulus* microfilarial prevalence in Togo's regions, comparing modelled projections with data from 400 endemic villages surveyed over time. We calculated the likelihood of EOT for each prefecture to determine which areas can begin stop-MDA surveys or may require ATS.

## Results
### Prevalence trends by region
Microfilarial prevalence trends were modelled for a total of 400 villages. Figures 2–6 present modelling results for the 140 OCP villages

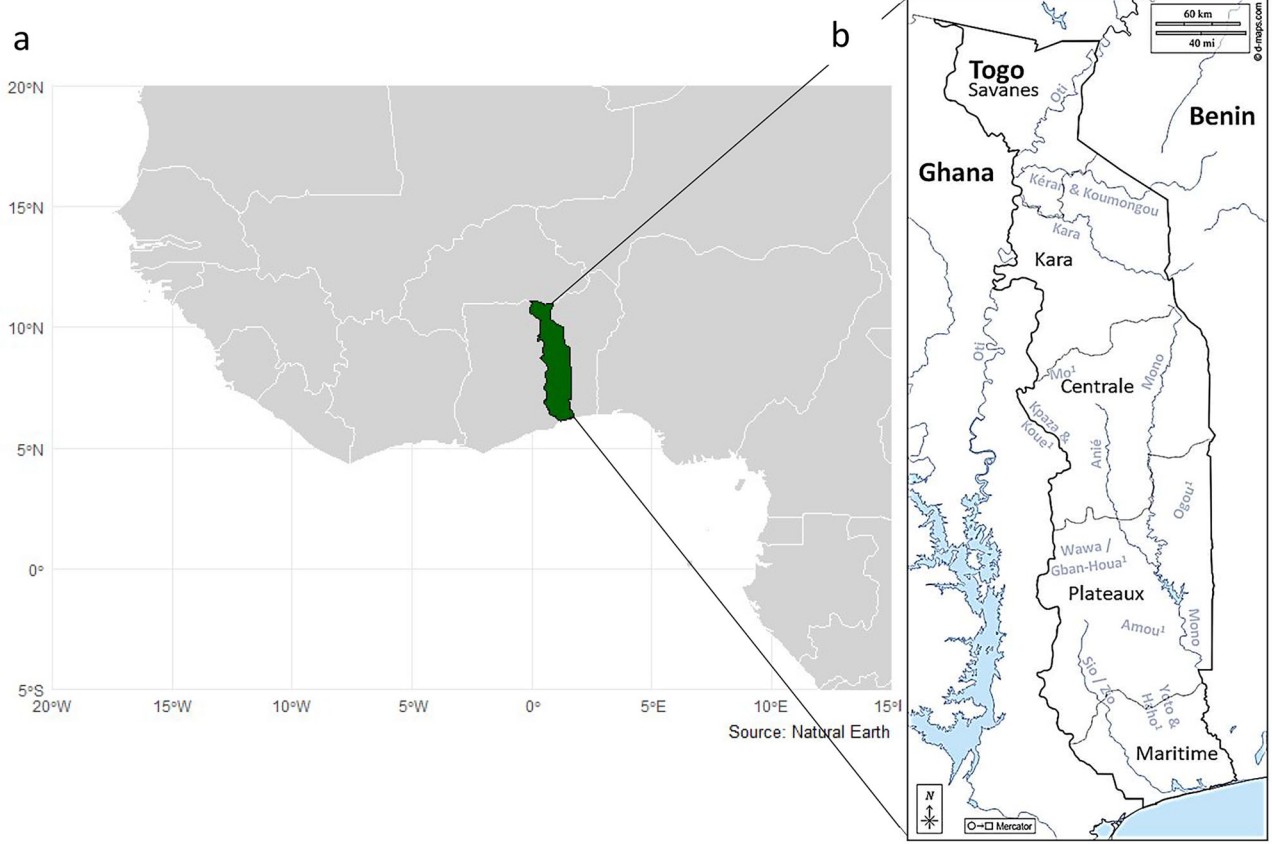

**Fig. 1 | Map of Togo. a** Location of Togo in West Africa. **b** Five regions of Togo and its main river basins. Togo regions are indicated by grey borders and names; main rivers are indicated by light blue and names. River basins with superscript 1 are named but not depicted. Mô is a tributary of the Oti River. Ogou and Amou are tributaries of the Mono River. Yoto and Haho flow towards the delta of the Maritime Region. Maps were drawn by the authors in R. Data sources: country outlines and coastlines in **a** from: Natural Earth (public domain, www.naturalearthdata.com); administrative boundaries in **b** from: GADM v4.0 accessed via the geodata R package (v0.6-2; CC BY 4.0, https://github.com/rspatial/geodata); river networks/basins in **b** from: HydroSHEDS (HydroRIVERS/HydroBASINS; CC BY 4.0, www.d-maps.com).

**Table 1 | Simulated duration of aerial vector control (VC) and ivermectin mass drug administration (MDA) in Togo regions**

| Intervention | Region and Special Intervention Zone (SIZ) status | | | | | | | |
| --- | --- | --- | --- | --- | --- | --- | --- | --- |
| | Savanes | | Kara | | Centrale | | Plateaux | Maritime |
| | SIZ | Non-SIZ | SIZ | Non-SIZ | SIZ | Non-SIZ | non-SIZ | non-SIZ |
| Start of VC | 1977[a] | 1977[a] | 1977[a] | 1977[a] | 1977 | 1989 | 1989 | 1988 |
| End of VC | 1993 | 1993 | 2007 | 1993 | 2007 | 2002 | 2002 | 2002 |
| Simulated duration of VC | 16 yr | 16 yr | 30 yr | 16 yr | 30 yr | 13 yr | 13 yr | 13 yr |
| Start of annual MDA[b] | 1991 | | | | | | | |
| Start of biannual MDA[c] | 2003 | NA | 2003 | NA | 2003 | NA | NA or 2014[d] | NA |
| Simulated end of MDA[e] | 2024, 2027 or 2030 | | | | | | | 2014 or 2020[f] |
| Simulated duration of MDA[f] | 12 yr annual; 21, 24 or 27 yr biannual | 33, 36 or 39 yr annual | 12 yr annual; 21, 24 or 27 yr biannual | 33, 36 or 39 yr annual | 12 yr annual; 21, 24 or 27 yr biannual | 33, 36 or 39 yr annual | 33, 36 or 39 yr annual and 10, 13 or 16 yr biannual | 23 or 29 yr annual |

[a]VC may have started in 1976 in some river basins[24,36].
[b]According to the data and literature, ivermectin MDA started earlier (1988–1990) in parts of Savanes and Kara (e.g., Bassar, Doufelgou, Kéran and Kozah prefectures)[24,48], but with poor coverage[49]. Some prefectures initiated MDA later (1992-1995). As most prefectures started MDA in 1991, this year was taken for the start of MDA in all simulations. NA = Not applicable (biannual MDA not implemented). Supplementary Information Table 2 presents prefecture-specific intervention details.
[c]In 2020, annual rather than biannual MDA was modelled in all prefectures across Togo due to the COVID-19 pandemic[45].
[d]In Plateaux four prefectures were switched to biannual MDA in 2014 because microfilarial prevalence in some villages was ≥5%[50].
[e]For the visualisation of infection trends, microfilarial prevalence dynamics were modelled until 2030, with the last simulated treatment round taking place in 2029. For the calculation of elimination of transmission (EOT) probabilities, ivermectin MDA was modelled to stop in 2024, 2027 or 2030.
[f]In Maritime, stop-MDA assessments were conducted in 2014–2017, indicating that it was possible to stop treatment in two prefectures. Subsequent stop-MDA assessments were performed in 2020–2023, showing that four prefectures were ready to stop treatment[27].

with recorded baseline microfilarial prevalence (BMP) estimates. Supplementary Figs. 8–14 present results for 252 villages without BMP estimates. (An additional 8, non-OCP villages also had BMP estimates.) Villages with BMP estimates were categorised into four endemicity levels: hypoendemic ( > 0% but <40%), mesoendemic ( ≥ 40% but <60%), hyperendemic ( ≥ 60% but <80%) and holoendemic ( ≥ 80%) microfilarial prevalence[20]. EPIONCHO-IBM[10] was implemented across four BMP values: 30%, 50%, 70% and 90% to capture these endemicity levels within Togo's five regions, SIZ status and intervention history (Table 1). Three ("minimal", "reference" and "enhanced") intervention scenarios were modelled (Table 2).

### Savanes
Villages with BMP estimates in Savanes prefectures were hypo- to mesoendemic in SIZ and meso- to hyperendemic in non-SIZ areas (Fig. 2). Prevalence decline was primarily driven by VC with enhanced (90%) or 100% efficacy (Fig. 2a–e), eventually leading to ≥90% EOT probability following initiation of MDA in hypo- to mesoendemic areas (Supplementary Table 12). Some SIZ villages lacking recorded BMP (Oti River Basin) had high prevalence, following hyperendemic simulation trends (Supplementary Fig. 8c). Non-SIZ villages lacking BMP estimates followed hypoendemic trends (Supplementary Fig. 9).

### Kara
Villages in Kara (all prefectures in SIZ) with recorded BMP estimates encompassed all endemicity levels (Fig. 3). The intervention scenarios best capturing infection trends were: minimal for hypoendemicity (Fig. 3a), reference for mesoendemicity (Fig. 3b), and minimal (Mô River Basin), reference (Kara River Basin) or enhanced (Kara River Basin) for hyperendemicity (Fig. 3c). Holoendemic villages aligned with the enhanced (Kéran River Basin) intervention scenario (Fig. 3d), with the model capturing the observed prevalence rebound after VC cessation. Two-fifths (30/74) of villages without BMP estimates followed hyper- to holoendemicity trends (Supplementary Information: Table 11, Fig. 10c–d).

### Centrale
Villages with recorded BMP in Centrale were hyperendemic in SIZ and ranged from hypo- to hyperendemic in non-SIZ areas. The SIZ hyperendemic villages (Mô River Basin) followed minimal, reference or enhanced intervention scenarios (Fig. 4a). Model projections for non-SIZ hypo- and mesoendemic villages (Fig. 4b, c) suggested a similar impact across intervention scenarios, likely because VC and MDA started roughly at the same time, reducing variability. Non-SIZ hyperendemic villages (Mono River Basin) followed the enhanced intervention scenario (Fig. 4d). Villages without recorded BMP ranged from hypo- to holoendemic (Supplementary Figs. 11–12). Nearly all SIZ villages (13/14) lacking BMP followed hyper- to holoendemic trends (Mô River Basin) (Supplementary Information: Table 11, Fig. 11).

### Plateaux
Villages in Plateaux (all non-SIZ) with recorded BMP were evenly distributed among hypo-, meso- and hyperendemicity (Fig. 5). Model outputs were similar for hypo- and mesoendemicity, as in Centrale (Fig. 5a–d). Prevalence trends in hyperendemic villages were mostly captured by the enhanced intervention scenario, with some following the minimal and reference scenarios (Fig. 5e, f, Mono River Basin). Several villages without BMP estimates followed hyperendemic trends (Supplementary Information: Table 11, Fig. 13). Prefectures continued with annual MDA or switched to biannual MDA in 2014 (Supplementary Table 2).

### Maritime
Villages in Maritime with recorded BMP were predominantly hypoendemic, and their modelled prevalence trends followed the

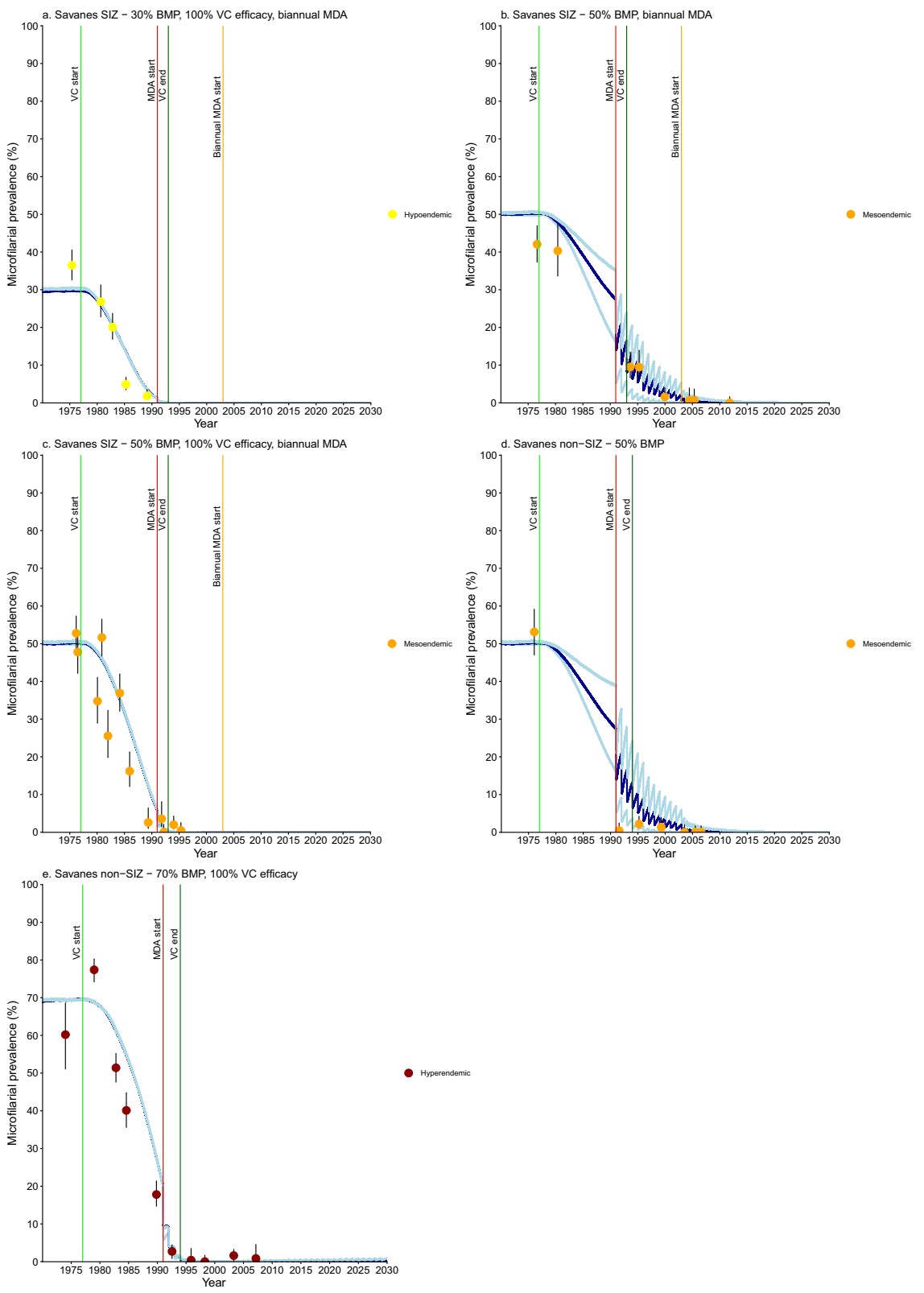

reference and enhanced intervention scenarios (Fig. 6a). The hyperendemic village depicted in Fig. 6b was consistent with the enhanced scenario. Villages without BMP estimates generally exhibited low endemicity, except in Yoto and possibly Avé prefectures, where some trends suggested hyperendemicity following the enhanced intervention scenario (Supplementary Information: Table 11, Fig. 14).

## Elimination probabilities

Supplementary Tables 12–17 present EOT probabilities per region, baseline endemicity, SIZ status and intervention scenario. In Savanes, all SIZ hypo- to (100% VC efficacy) hyperendemic villages and nearly all (86%) non-SIZ villages are projected to have reached ≥90% EOT probability by 2024. However, in the northwestern part of Savanes non-SIZ, surveys conducted in early to mid-1970s (not in the OCP

**Fig. 2 | *Onchocerca volvulus* microfilarial prevalence trends simulated using EPIONCHO-IBM (until 2030) for Savanes Region villages with recorded pre-control baseline microfilarial prevalence (BMP) estimates, vector control (VC) and ivermectin mass drug administration (MDA).** For Special Intervention Zone (SIZ): **a** hypoendemic village (*n* = 1) with simulated 30% BMP, 100% VC efficacy and biannual MDA. **b** mesoendemic village (*n* = 1) with simulated 50% BMP and biannual MDA. **c** mesoendemic villages (*n* = 2) with simulated 50% BMP, 100% VC efficacy and biannual MDA. For Non-Special Intervention Zone (non-SIZ): **d** mesoendemic village (*n* = 1) with simulated 50% BMP. **e** hyperendemic villages (*n* = 2) and 70% BMP with 100% VC efficacy. Yellow circles represent microfilarial prevalence of hypoendemic villages, orange circles indicate microfilarial prevalence of mesoendemic villages, and brown circles denote microfilarial prevalence of hyperendemic villages. Error bars are 95% (Wilson score) confidence intervals (95% CIs). For each BMP setting and intervention scenario, the average of 100 model repeats was used to calculate the mean microfilarial prevalence dynamics (blue lines). Dark blue lines represent the reference scenario; light blue lines above and below dark blue lines indicate the minimal and enhanced scenarios, respectively. Vertical coloured lines indicate: start of VC (light green); start of annual MDA (red); end of VC (dark green); start of biannual MDA (orange). Further information about the individual villages plotted here can be found in Supplementary Table 10. Source data and code are available[47].

**Table 2 | Vector control (VC) efficacy, therapeutic coverage (of total population) and proportion of systematic non-adherence (SNA) for ivermectin mass drug administration (MDA) for the three intervention scenarios simulated for Togo**

| Scenario | VC efficacy[a] | Ivermectin MDA therapeutic coverage | | | SNA |
|---|---|---|---|---|---|
| | | 1991–1995 | 1996–2001 | 2002–2030 | |
| Minimal (upper uncertainty bound) | 60% | 50% | 65% | 65% | 5.0% |
| Reference (average) | 75% | 50% | 65% | 75% | 2.5% |
| Enhanced (lower uncertainty bound) | 90% | 65% | 75% | 80% | 1.0% |

[a]For Savanes, 100% VC efficacy simulations were also run for the three scenarios[44]. See Supplementary Information Text 4 for further details, and Supplementary Tables 5–9 for reported coverage of total population.

database), indicated baseline hyperendemicity (Supplementary Fig. 1). As non-SIZ areas of Savanes did not receive biannual MDA and VC ceased in 1993 (Table 1), the projected EOT probabilities are <5%. SIZ villages without BMP estimates following hyperendemic trends are projected to have <90% probability of reaching EOT by 2024. Extending biannual MDA to 2027 or 2030 in putative hyperendemic villages does not improve their EOT probabilities, remaining at <5%, 20–59%, and 60–89%, under minimal, reference, and enhanced interventions, respectively (Supplementary Table 12).

In Kara and Centrale, most hypo- and mesoendemic villages (regardless of SIZ status and intervention scenario) are projected to have reached ≥90% EOT probability by 2024. Some non-SIZ mesoendemic villages of Centrale following the minimal scenario and annual MDA would only reach 60–89% EOT probability, even with MDA extended to 2030. Hyperendemic villages following the enhanced scenario are projected to have reached ≥90% EOT probability by 2024. By contrast, those following the minimal or reference scenarios would have 5–19% or 60–89% EOT probability by 2024, respectively. If treatment continues until 2030, the former's EOT probability would increase to 20–59%. As in Savanes, holoendemic villages have <5% EOT probability (Supplementary Tables 13–14).

In Plateaux, owing to its lower BMP compared to Kara and Centrale, VC started later and not all prefectures adopted biannual MDA. Hypoendemic villages (irrespective of treatment frequency) and mesoendemic villages following reference and enhanced scenarios with annual or biannual MDA, are projected to have reached ≥90% EOT probability by 2024. The same applies to mesoendemic villages following the minimal scenario under biannual MDA. Mesoendemic villages following the minimal scenario and hyperendemic villages following the enhanced scenario under annual MDA, would have had 60–89% EOT probability by 2024. Hyperendemic villages following minimal and reference scenarios under annual MDA would only reach <5% EOT probability by 2024 or 2030. Those following minimal and reference scenarios under biannual MDA would have reached, respectively, <5% and 5–19% EOT by 2024, with the latter increasing to 20–59% if biannual MDA were extended until 2030. By contrast, hyperendemic villages following the enhanced scenario and already under biannual MDA could reach ≥90% EOT if biannual treatment continues until 2030 (Supplementary Table 15).

In Maritime, only annual MDA had been implemented by 2018. Being the least endemic region, all hypoendemic villages, and those mesoendemic villages following the reference and enhanced scenarios, are projected to have reached ≥90% EOT probability regardless of whether treatment ceased in 2014 or 2020. However, in one confirmed (with) and several putative (without BMP) hyperendemic villages, the EOT probability is 20–59% in 2020. Extending treatment in such villages to 2024 or 2030 would increase it to 60–89% (Supplementary Tables 16–17). Supplementary Tables 18–29 list villages (with and without BMP estimates) by region, SIZ status, and river basin for which EPIONCHO-IBM projects EOT probabilities <90% if MDA stops in 2027. Supplementary Tables 32–34 present prefecture-level likelihood of reaching EOT when simulating that ivermectin MDA stops in 2024, 2027 or 2030. Fig. 7 illustrates the (categorical) likelihood of reaching EOT if ivermectin MDA stops in 2027 across Togo. Supplementary Table 35 summarises current control strategies and recommendations by prefecture.

## Discussion

Modelling analyses of detailed spatiotemporal *O. volvulus* prevalence data from the OCP and other sources provided a unique opportunity to quantify the combined impact of VC and ivermectin MDA in a former OCP country. The epidemiology of onchocerciasis in Togo has changed profoundly over nearly 50 years of intervention, with some prefectures on the verge of reaching EOT and others projected not to reach EOT with current strategies. Our results illustrate the power of mathematical modelling in evaluating past, current and future epidemiology of onchocerciasis, to assist programmes in decision-making and resource allocation to maximise their chances of reaching the 2030 elimination goals.

We identified regional epidemiological patterns, long-term prevalence declines, and subsequent increases in some cases. In Savanes non-SIZ, (hypo- to hyperendemic) villages with 90-100% VC efficacy had reached a very low microfilarial prevalence by the time MDA started. This contrasts with other regions where VC was likely less impactful, underscoring the need for combining VC and MDA. After VC cessation, hypo- to mesoendemic villages performed well under MDA. Annual MDA sustained previous gains in hyperendemic villages, but only biannual MDA led to further prevalence declines. Holoendemic

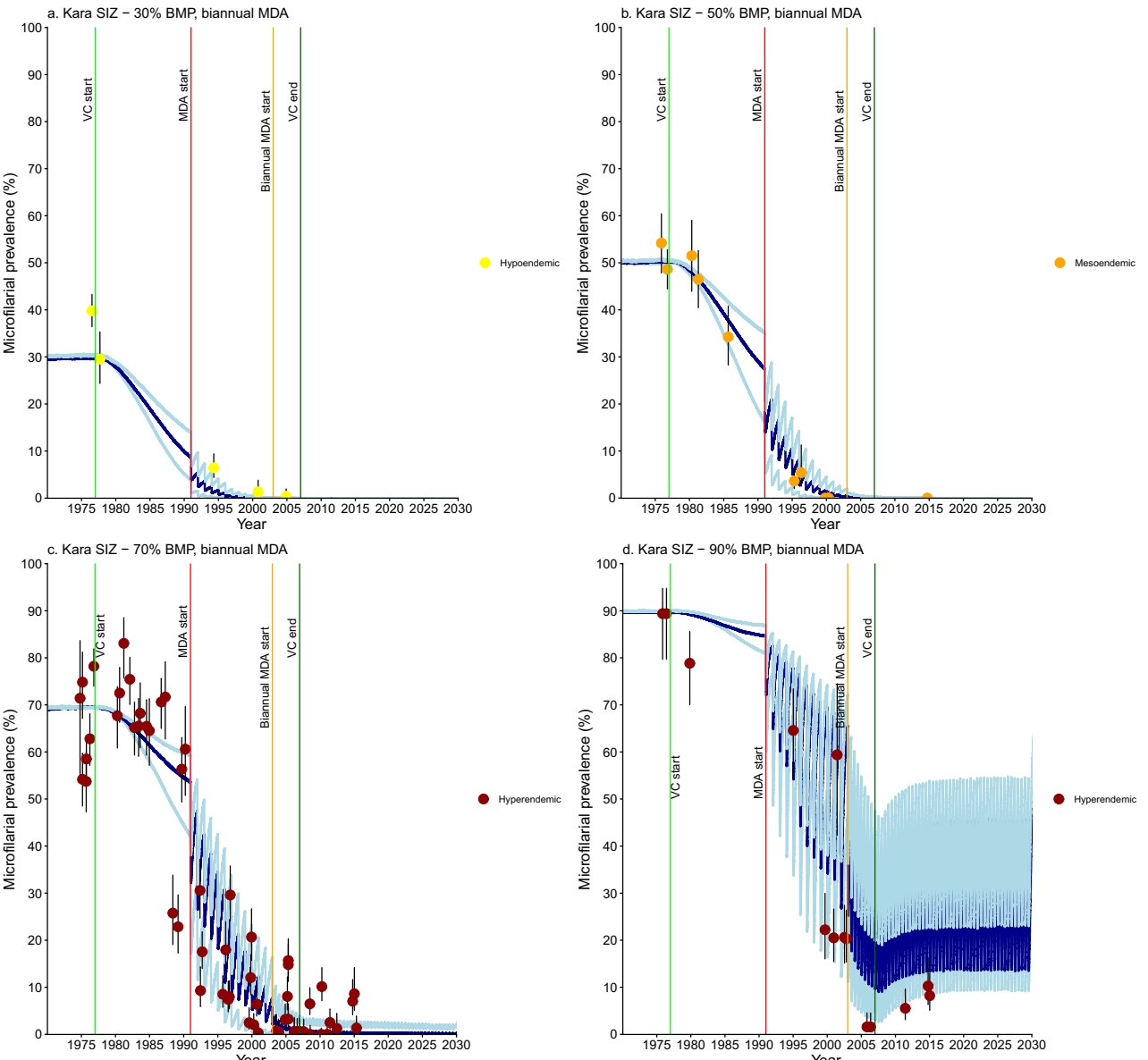

**Fig. 3 | *Onchocerca volvulus* microfilarial prevalence trends simulated using EPIONCHO-IBM (until 2030) for Kara Region villages with recorded pre-control baseline microfilarial prevalence (BMP) estimates, vector control (VC) and ivermectin mass drug administration (MDA).** For Special Intervention Zone (SIZ, all villages): **a** hypoendemic villages (*n* = 2) with simulated 30% BMP and biannual MDA. **b** mesoendemic villages (*n* = 2) with simulated 50% BMP and biannual MDA. **c** hyperendemic villages (*n* = 6) with simulated 70% BMP and biannual MDA. **d** holoendemic villages (*n* = 2) with simulated 90% BMP and biannual MDA. Yellow circles represent microfilarial prevalence of hypoendemic villages, orange circles indicate microfilarial prevalence of mesoendemic villages, and brown circles denote microfilarial prevalence of hyper- and holoendemic villages. Error bars are 95% (Wilson score) confidence intervals (95% CIs). For each BMP setting and intervention scenario, the average of 100 model repeats was used to calculate the mean microfilarial prevalence dynamics (blue lines). Dark blue lines represent the reference scenario; light blue lines above and below dark blue lines indicate the minimal and enhanced scenarios, respectively. Vertical coloured lines indicate: start of VC (light green); start of annual MDA (red); end of VC (dark green); start of biannual MDA (orange). Further information about the individual villages plotted here can be found in Supplementary Table 10. Source data and code are available[47].

villages experienced prevalence increases after VC cessation, in both data and model outputs, even with biannual MDA.

According to EPIONCHO-IBM, achieving ≥90% EOT probability by 2024 in hypo- and mesoendemic areas nationwide seems feasible, and stop-MDA surveys could start if not already underway. Conversely, modelled prevalence in areas with hyper- and holoendemic villages, such as the Oti River Basin (Savanes), and the Kéran and Mô River Basins (Kara and Centrale), declines to a pseudo-equilibrium (a seemingly steady state of infection maintained by the opposing effects of treatment and microfilarial prevalence recovery under conditions of intense inter-treatment transmission) since 2007, and are unlikely to achieve EOT under the current (biannual) MDA strategy, even if continued until 2030. Infection prevalence of 0.1–1.0% in blackflies was found for these river basins (2015), and of 0.5–0.8% in Mô (2018–2019), indicating active transmission[24,25]. In non-SIZ (northwestern) Savanes, where VC started and ended early[15], surveyed villages (with or without BMP estimates) are scarce in our database. However, based on surveys indicating hyperendemicity in the 1970s, EOT likelihood in Tandjouaré and Tône prefectures is <90%. Stop-MDA surveys in 2022 indicated some villages of potential concern in these prefectures[26].

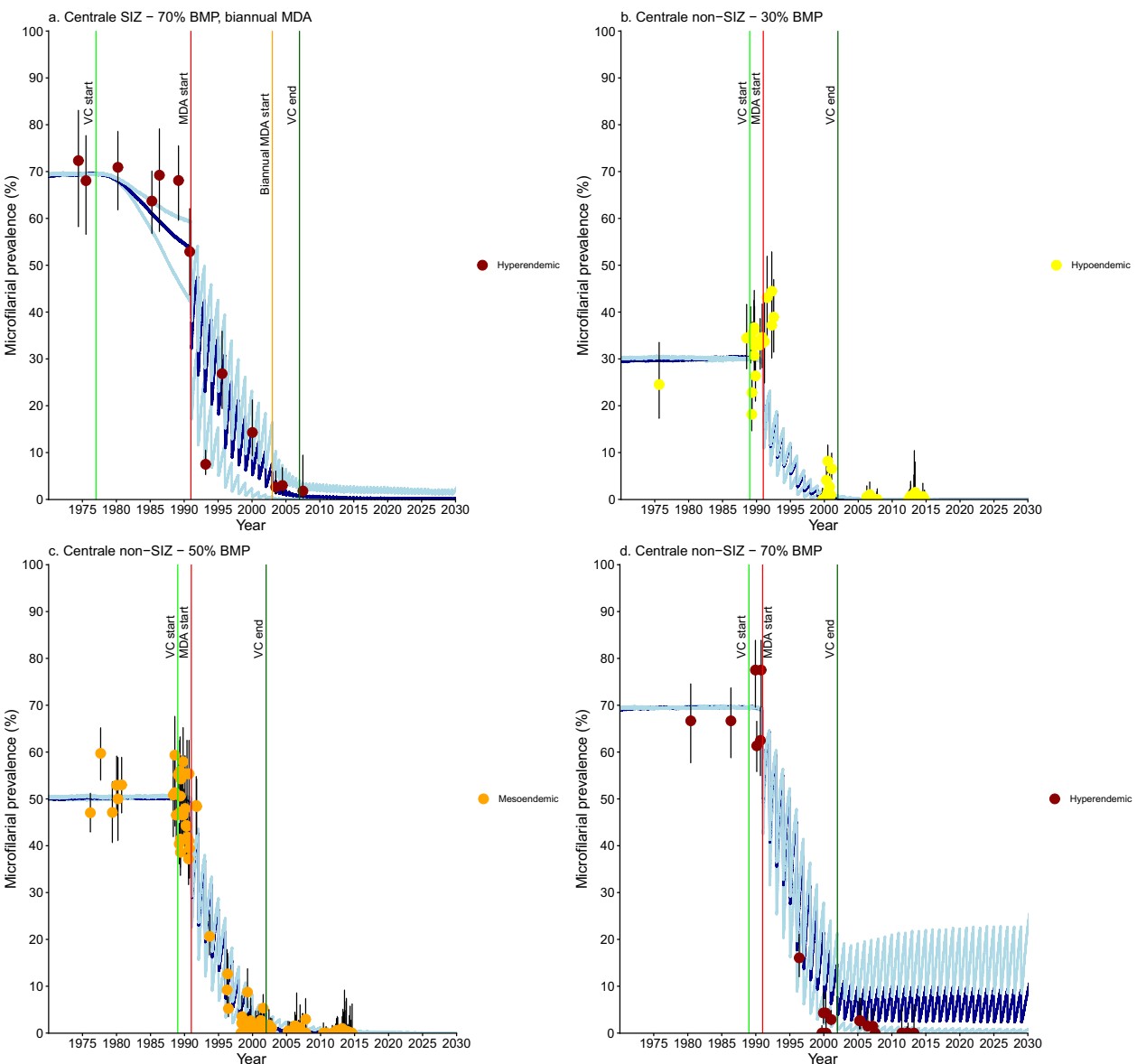

**Fig. 4 | *Onchocerca volvulus* microfilarial prevalence trends simulated using EPIONCHO-IBM (until 2030) for Centrale Region villages with recorded pre-control baseline microfilarial prevalence (BMP) estimates, vector control (VC) and ivermectin mass drug administration (MDA).** For Special Intervention Zone (SIZ): **a** hyperendemic villages (*n* = 2) with 70% BMP and biannual MDA. For Non-Special Intervention Zone (non-SIZ): **b** hypoendemic villages (*n* = 13) with 30% BMP. **c** mesoendemic villages (*n* = 26) with 50% BMP. **d** hyperendemic villages (*n* = 5) with 70% BMP. Yellow circles represent microfilarial prevalence of hypoendemic villages, orange circles indicate microfilarial prevalence of mesoendemic villages, and brown circles denote microfilarial prevalence of hyperendemic villages. Error bars are 95% (Wilson score) confidence intervals (95%CIs). For each BMP setting and intervention scenario, the average of 100 model repeats was used to calculate the mean microfilarial prevalence dynamics (blue lines). Dark blue lines represent the reference scenario; light blue lines above and below dark blue lines indicate the minimal and enhanced scenarios, respectively. Vertical coloured lines indicate: start of VC (light green); start of annual MDA (red); end of VC (dark green); start of biannual MDA (orange). Further information about the individual villages plotted here can be found in Supplementary Table 10. Source data and code are available[47].

Whilst most of Maritime would likely have reached ≥90% EOT probability by 2020, some villages without BMP estimates, mostly in Yoto Prefecture, followed hyperendemic trends with moderate (60–89%) EOT probability. Active transmission was confirmed in Yoto during the 2020–2023 stop-MDA survey, prompting focal biannual MDA[27].

**Limitations**. Although baseline annual biting rates (ABRs) were reduced during VC by its assumed efficacy, they were modelled as bouncing back to pre-control levels one year after VC cessation[28]. This conservative assumption is supported by several entomological studies conducted in Togo and other OCP countries. In the Mô River Basin, ABRs in 2015–2019 (12,000–16,000 bites/person/year)[24,25] were comparable to pre-VC values (16,000–47,000). In areas of Burkina Faso bordering with Togo, ABRs returned to or surpassed baseline levels within two years of stopping VC (Loaba: 8617 compared to 6090; Ziou Zabré: 30,739 compared to 11,879)[7]. In Niger, the Goulbi River Basin experienced a bounce-back to 14,000 bites/person/year within one year after ceasing vector control, compared to baseline values of 14,350[29]. Even in areas experiencing deforestation as in the Wawa River in Ghana (a water course shared with Plateaux), biting rates rebounded to pre-control levels after VC ended, reaching 2400–4000 compared to 2800–4050 bites/person/month[30]. Notwithstanding, deforestation, particularly in western Plateaux and southern Centrale[31], may have led to secular changes in vector density[32] not considered in the model.

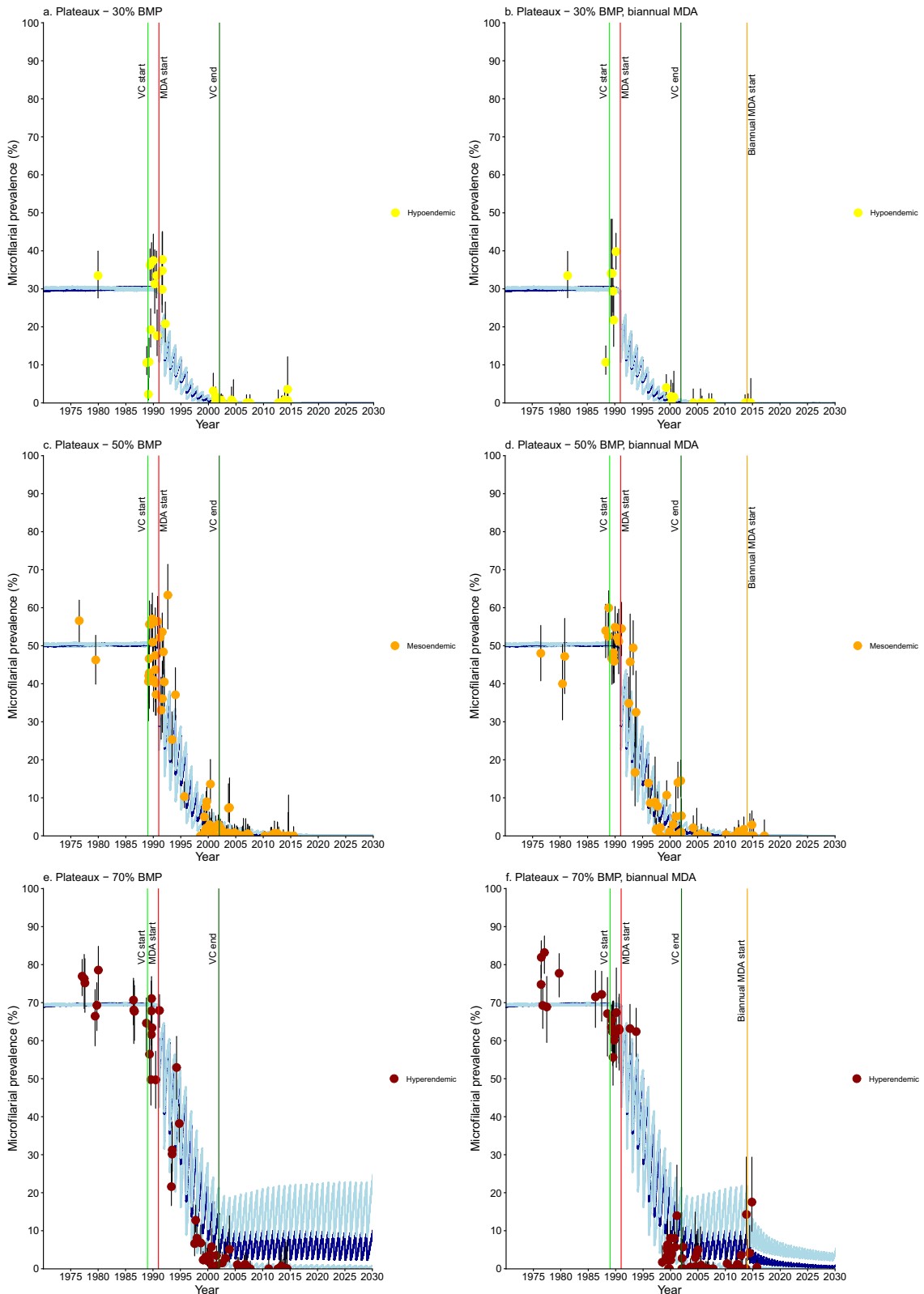

Future modelling studies should account for secular trends in ABRs when substantial changes in transmission conditions have been documented[33]. Although we modelled information from 400 villages, there may be many others, certainly those with >2000 people, for which we have no epidemiological data as they were only incorporated into the NOCP in 2018[23]. Because we may not have fully captured within-region heterogeneity, our prefecture-wide EOT likelihood must be interpreted with caution. Also, EPIONCHO-IBM models closed populations, not accounting for movement of people and/or flies between villages or cross-border migration that could jeopardise EOT by re-introduction of infection from less-well controlled areas[26]. Recently, EPIONCHO-IBM has been used to model the effects, on introduction or re-introduction of infection, of an influx of immigrants with varying worm burdens arriving from an area with ongoing

**Fig. 5 | *Onchocerca volvulus* microfilarial prevalence trends simulated using EPIONCHO-IBM (until 2030) for Plateaux Region villages (all non-Special Intervention Zone) with recorded pre-control baseline microfilarial prevalence (BMP) estimates, vector control (VC) and ivermectin mass drug administration (MDA). a** hypoendemic villages (*n* = 12) with simulated 30% BMP and annual MDA. **b** hypoendemic villages (*n* = 4) with 30% BMP and biannual MDA. **c** mesoendemic villages (*n* = 15) with 50% BMP and annual MDA. **d** mesoendemic villages (*n* = 12) with 50% BMP and biannual MDA. **e** hyperendemic villages (*n* = 12) with 70% BMP and annual MDA. **f** hyperendemic villages (*n* = 13) with 70% BMP and biannual MDA. Yellow circles represent microfilarial prevalence of hypoendemic

villages, orange circles indicate microfilarial prevalence of mesoendemic villages, and brown circles denote microfilarial prevalence of hyperendemic villages. Error bars are 95% (Wilson score) confidence intervals (95%CIs). For each BMP setting and intervention scenario, the average of 100 model repeats was used to calculate the mean microfilarial prevalence dynamics (blue lines). Dark blue lines represent the reference scenario; light blue lines above and below dark blue lines indicate the minimal and enhanced scenarios, respectively. Vertical coloured lines indicate: start of VC (light green); start of annual MDA (red); end of VC (dark green); start of biannual MDA (orange). Further information about individual villages plotted here can be found in Supplementary Table 10. Source data and code are available[47].

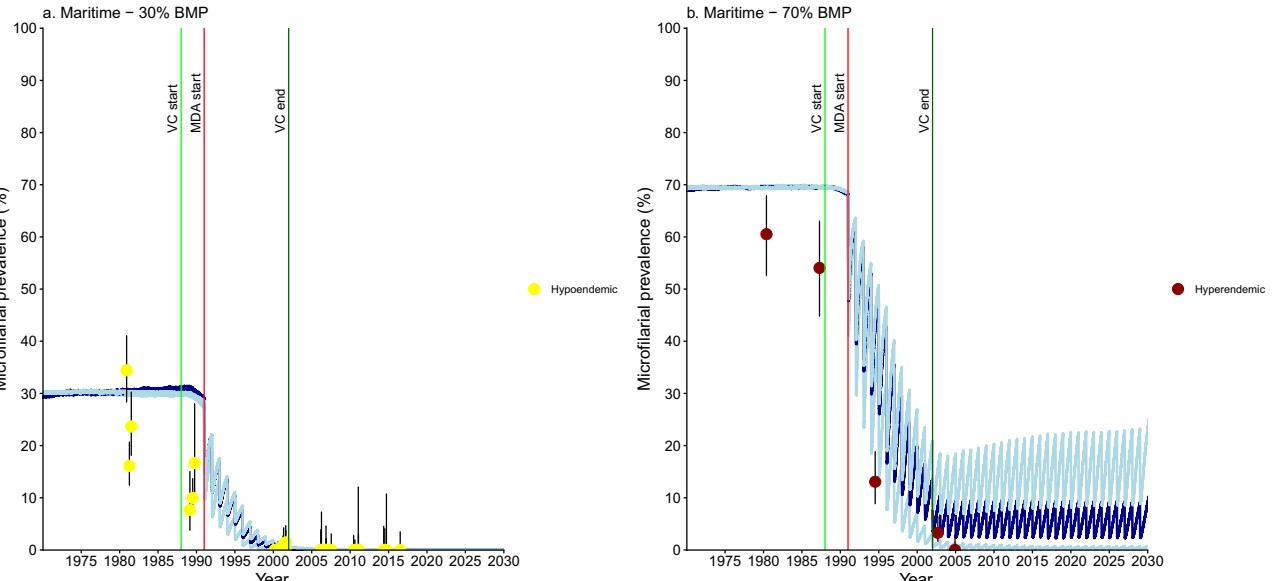

**Fig. 6 | *Onchocerca volvulus* microfilarial prevalence trends simulated using EPIONCHO-IBM (until 2030) for Maritime Region villages (all non-Special Intervention Zone) with recorded pre-control baseline microfilarial prevalence (BMP) estimates, vector control (VC) and ivermectin mass drug administration (MDA). a** hypoendemic villages (*n* = 6) with simulated 30% BMP. **b** hyperendemic village (*n* = 1) with 70% BMP. Yellow circles represent microfilarial prevalence of hypoendemic villages, and brown circles denote microfilarial prevalence of a hyperendemic village. Error bars are 95% (Wilson score) confidence

intervals (95%CIs). For each BMP setting and intervention scenario, the average of 100 model repeats was used to calculate the mean microfilarial prevalence dynamics (blue lines). Dark blue lines represent the reference scenario; light blue lines above and below dark blue lines indicate the minimal and enhanced scenarios, respectively. Vertical coloured lines indicate: start of VC (light green); start of annual MDA (red); end of VC (dark green). Further information about the individual villages plotted here can be found in Supplementary Table 10. Source data and code are available[47].

transmission into an onchocerciasis-free setting with local blackfly populations[34].

In conclusion, Togo has made considerable progress towards onchocerciasis EOT owing to VC and ivermectin MDA, switching to biannual frequency where necessary. However, areas with confirmed or putative high baseline endemicity pose challenges to achieving nation-wide EOT. EPIONCHO-IBM has proven its usefulness in interpreting epidemiological data, supporting decisions on stop-MDA surveys (e.g., in hypo- to mesoendemic areas with ≥15 years of high-coverage MDA and/or biannual treatment). In hyper- and holendemic areas with low EOT probabilities (e.g., in Kara and Centrale), the model suggests that ATS[21] should be considered. In particular, biannual moxidectin MDA supplemented, if feasible, by "slash-and-clear" VC would be beneficial[22,35]. EPIONCHO-IBM could be used in other former OCP countries to inform policy decisions towards the 2030 elimination goals.

## Methods
### Ethics statement
This study analysed de-identified, aggregate village-level microfilarial prevalence data collected in Togo between 1970 and 2017 by the OCP,

and national and partner programmes, which obtained ethical approval from their respective boards at the time of the surveys (e.g., from the Togo Bioethics Committee for Research in Health (Comité de Bioéthique pour la Recherche en Santé), with authorization and approval granted by the Togo Ministry of Health)[24,25]. In the OCP, a memorandum of agreement was signed covering all operational issues as well as clearance for epidemiological surveys. A committee consisting of OCP Chief of Units ensured that work plans and methodology were correctly followed by field technicians. Communities were free to participate in the taking of skin snip samples[5]. For this study, no human participants were recruited and no individual-level identifiable data were used; therefore, no institutional review board approval and informed consent were required for the present (modelling) analysis.

### Prevalence data
Cross-sectional surveys (1970–2017) provided microfilarial prevalence data for 400 endemic villages (Supplementary Information: Text 2, Table 1, Figs. 3–5). Villages were considered to have recorded baseline endemicity estimates if surveyed for *O. volvulus* microfilarial prevalence before the start of ivermectin MDA or <3 years after starting VC[7,36] (*n* = 148 with BMP estimates; 140 in the OCP database). Other

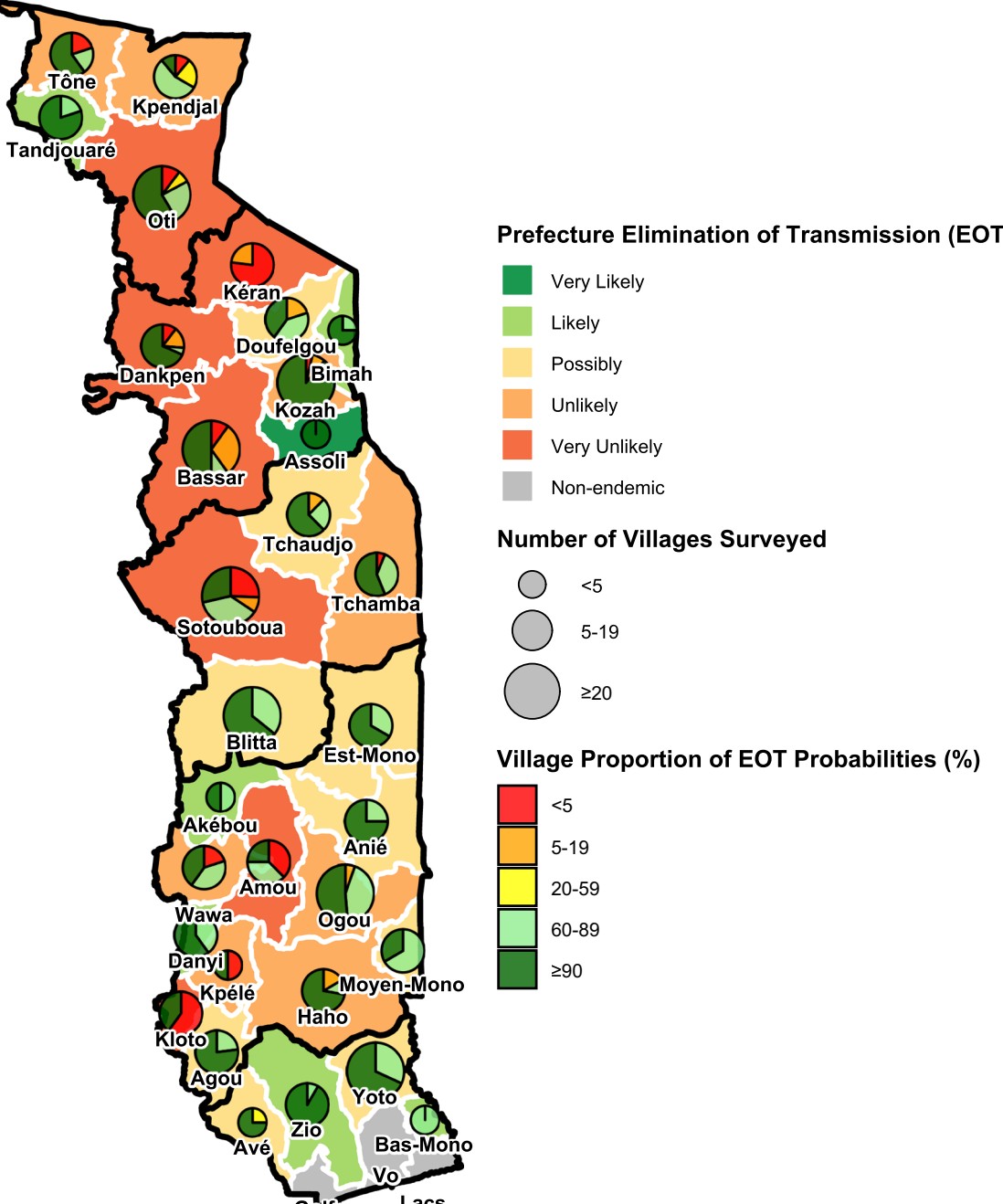

**Fig. 7 | Categorical likelihood of elimination of onchocerciasis transmission (EOT) in Togo's prefectures if ivermectin mass drug administration were stopped in 2027, according to EPIONCHO-IBM projections.** Details of the calculation of the prefecture-wide (joint) EOT probabilities are given in Supplementary Text 8. Categories are defined as: Very likely (≥ 90%); Likely (50−89%); Possibly (5−49%); Unlikely (0.01- < 5%), and Very unlikely (< 0.01%). The pie-charts indicate the proportion of the total number of villages surveyed in each prefecture that are projected to reach <5%, 5–19%, 20–59%, 60–89% or ≥90% EOT probability, with the size of the pie-charts reflecting the number of villages (exact numbers are given in Supplementary Table 33; detailed information on those villages with projected EOT probability <90% is presented in Supplementary Tables 18–29). Black borders indicate regions (see Fig. 1); white borders correspond to prefectures. The map for Togo (with regions and prefectures) was drawn by the authors in R using the geodata package (version 0.6-2; CC BY 4.0, https://github.com/rspatial/geodata). Source data and code are available[47].

villages lacked baseline assessments (*n* = 252) but had surveys conducted after the start of interventions (Supplementary Table 1). We used crude rather than age- and sex-standardised microfilarial prevalence because the latter was missing in 12% of the surveys, and there was a 0.99 Pearson's correlation coefficient between crude and standardised prevalence (Supplementary Fig. 6)[36]. Wilson-score 95% confidence intervals (95% CIs)[37] were calculated for each prevalence estimate.

Four endemicity levels (hypo-, meso-, hyper- and holoendemic)[20] were used to categorise villages with BMP estimates, for which 30%, 50%, 70% or 90% BMP values were modelled to capture their endemicity levels. For villages without recorded BMP, all four endemicity levels were simulated to identify their most likely initial endemicity category according to modelled microfilarial prevalence trajectories (Supplementary Table 1). Baseline annual biting rates (ABRs, bites/person/year) were estimated by interpolating the relationship between

microfilarial prevalence and ABR using the EPIONCHO-IBM transmission model[10], generating ABR = 290 (for 30%), 615 (50%), 2,200 (70%), and 60,000 (90%) (Supplementary Information: Text 3, Tables 3–4).

## EPIONCHO-IBM

EPIONCHO-IBM is a stochastic, individual-based model simulating *O. volvulus* infection dynamics in a closed population (village)[10] (500 individuals for this work). It tracks the number of (male and female) adult worms and microfilarial load in each human host over time, and the mean number of infective, L3 larvae per blackfly vector. Adult worm and microfilarial mortality rates are parasite-age dependent, and adult female worm fecundity decreases with worm age[10]. Human exposure is age- and sex-dependent[38], and overdispersed among individuals following a gamma distribution with shape and rate parameter $k_E$ (=0.3 for this work)[10]. Parasite population abundance is regulated by density-dependent processes within humans and vectors[39], which contribute to endemic stability and intervention resilience[10,40]. A description of the model is provided in Hamley et al.[10] (code available at: https://github.com/mrc-ide/EPIONCHO.IBM)[41].

## Modelling interventions and scenarios

EPIONCHO-IBM was implemented across the four endemicity levels aforementioned within Togo's five regions, considering their SIZ status and intervention history (Table 1). Supplementary Table 2 provides intervention details at prefecture level. Modelled interventions comprised VC and ivermectin MDA. Ivermectin MDA was modelled by incorporating microfilaricidal and embryostatic effects[42], and a permanent sterilising effect on adult female worms[43]. Therapeutic coverage (proportion of individuals receiving ivermectin at each round in the total population) was simulated as the mean treatment probability in any treatment round. A fixed proportion of systematic nonadherence (SNA) was used to represent eligible individuals never receiving treatment[10]. VC was simulated by reducing ABR based on assumed efficacy for the entire larviciding duration. ABR values were assumed to return to initial levels one year after VC cessation[28]. Supplementary Tables 5–9 provide reported coverage data by region and year.

In addition to the three intervention scenarios described in Table 2 (and Supplementary Text 4), a 100% VC efficacy[44] was simulated in Savanes, varying therapeutic coverage and SNA as per the three main scenarios. Biannual MDA started in 2003 or 2014 in some regions (Table 1). In 2020, MDA was modelled annually nationally due to the COVID-19 pandemic[45]. Microfilarial prevalence trends were simulated until 2030, with the last (annual or biannual) treatment in 2029.

A total of 100 model repeats were run for each of the three intervention scenarios, four endemicity levels, five regions and SIZ status. The mean of the 100 runs yielded mean microfilarial prevalence dynamics over time. Prevalence trends were visualized using the "minimal" and "enhanced" scenarios as the upper and lower uncertainty bounds, with "reference" scenario representing the average. Simulated trends are presented alongside survey prevalence estimates with 95% CIs per region and SIZ status to illustrate infection trends and compare model outputs with observations (village surveys). The proportion of the population examined for *O. volvulus* skin microfilariae decreased from >80% at the beginning of the OCP to <70% between 2006 and 2015 (Supplementary Information: Text 5, Fig. 7). Supplementary Table 10 lists the villages with recorded BMP.

To identify the intervention scenarios followed by villages lacking BMP estimates, we used the (median) Mean Squared Error (MSE) between observed and simulated microfilarial prevalence trends across model repeats, with the smallest MSE values indicating the best-fit scenarios (Supplementary Information: Text 6, Table 11).

## Elimination probabilities

For simulation of EOT probabilities, MDA frequencies from 2018 (annual or biannual, depending on prefecture) were used. The probability of reaching EOT for each scenario, region and SIZ status was calculated as the percentage of 100 model runs yielding zero microfilarial prevalence 50 years after stopping MDA[35] in 2024, 2027 or 2030. For Maritime, where MDA ended in some prefectures in 2014 and in all by 2020, simulations ceasing MDA in 2014 or 2020 were performed (Table 1 and Supplementary Table 2). Five EOT probability categories were defined: <5%, 6–19%, 20–59%, 60–89% and ≥90%. Supplementary Information: Text 8, Tables 30–31 present the rationale and calculations of prefecture-wide likelihoods (joint probabilities) of achieving EOT.

We followed the five principles of the NTD Modelling Consortium regarding Policy-Relevant Items for Reporting Models in Epidemiology of NTDs (PRIME-NTD)[46] (Supplementary Information: Text 8, Table 36).

## Reporting summary

Further information on research design is available in the Nature Portfolio Reporting Summary linked to this article.

## Data availability

All the information used for the analyses presented here is contained in the figures, tables and Supplementary Information of this paper, and in the (publicly available) epidemiological database (https://doi.org/10.1371/journal.pntd.0012312.s001; https://doi.org/10.1371/journal.pntd.0012312.s003) found in Vinkeles Melchers et al[15].

## Code availability

The EPIONCHO-IBM model code is available at: https://github.com/mrc-ide/EPIONCHO.IBM[41]. The following link provides the branch for the modelling analysis presented in the paper: https://github.com/mrc-ide/EPIONCHO.IBM/tree/togo-elimination-analysis. The R code to reproduce the modelling analysis and figures (https://doi.org/10.5281/zenodo.17351356) can be found in Amaral & Basáñez[47].

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

## Acknowledgements

L.-J.A. was funded by La Caixa Foundation (grant B005782). M.W. and M.-G.B. acknowledge funding by the Bill & Melinda Gates Foundation through the NTD Modelling Consortium (grants OPP1184344 and INV-030046). M.-G.B. acknowledges funding from the MRC Centre for Global Infectious Disease Analysis (grant MR/X020258/1), funded by the UK Medical Research Council (MRC). This UK-funded award is carried out in the frame of the Global Health EDCTP3 Joint Undertaking. Special thanks go to Dr Paul Cantey for his introduction to the Togo team, and to Prof. Robert Colebunders for his support. Dr Natalie Vinkeles Melchers provided useful advice on the assumptions used for the control interventions implemented in Togo that informed Table 1. We also acknowledge Dr Philip Milton for his guidance during the early stages of the work, and Mr Aditya Ramani for calculating the range of annual biting rates for the holoendemic settings presented in Supplementary Information and advice on the Mean Square Error calculations and the preparation of the GitHub link to the code contained in the Zenodo link. This paper is dedicated to the memory of Prof. Yao Kassankogno whose contribution to data availability was invaluable. The funders had no role in study design, data collection and analysis, decision to publish, or preparation of the manuscript. The corresponding authors had final responsibility for the decision to submit for publication.

## Author contributions

Conceptualization: L.-J.A., M.-G.B. Data exploration: L.-J.A., J.I.D.H. Data Curation: J.-L.A. Formal analysis: L.-J.A. Investigation and methodology: L.-J.A., J.I.D.H., M.W., M.-G.B. Software: J.I.D.H. Resources: L.-J.A., R.N.B., A.S., M.-G.B. Visualization: L.-J.A., M.W., M.-G.B. Supervision: M.W., M.-G.B. Funding acquisition: L.-J.A., M.-G.B. Project administration: R.N.B., M.-G.B. Writing—original draft: L.-J.A., M.-G.B. Writing—review and editing: L.-J.A., R.N.B., A.S., M.-D.M., K.P., I.G.T., S.A., M.D., P.G., J.I.D.H., M.W., M.-G.B.

## Competing interests

The authors declare no competing interests.
