## [Peer Review file · Nature Communications]

Reaching Elimination of Onchocerciasis Transmission with Long-term Vector Control and Ivermectin Treatment in Togo

Corresponding Author: Professor María-Gloria Basáñez

Version 0:

Reviewer comments:

Reviewer #1

(Remarks to the Author)

This manuscript presents a comprehensive evaluation of onchocerciasis elimination efforts in Togo through the application of the EPIONCHO-IBM transmission model. Using detailed village-level epidemiological data spanning nearly five decades, the authors simulate the impact of historical and current interventions across different endemicity settings and regions. The manuscript is a strong and policy-relevant contribution to the field of NTDs and offers critical insights into the heterogeneity of progress toward elimination of transmission (EOT) in Togo. The manuscript offers novel insights by combining long-term prevalence data with a validated individual-based model (EPIONCHO-IBM). It stands out for the breadth of the dataset (400 villages) and the policy utility of projecting EOT probabilities at sub-national level. The classification of regions into stop-MDA-ready and ATS-needing zones is especially useful for national programs and aligns well with WHO guidelines. Regarding model implementation, the inclusion of the EPIONCHO-IBM code link and open-access data enhances reproducibility. The authors also align their reporting with the PRIME-NTD principles (Supplementary Material 1, Text S6, p. 31), which is commendable.

Major comments

1. While the authors justifiably assume that annual biting rates (ABRs) return to pre-intervention levels post-vector control (post-VC) (Methods, p. 8), this assumption may underestimate the sustained impact of ecological or human-induced changes (e.g., deforestation, population movements, changes on human habits leading to decline in human-vector contact, etc.). A brief sensitivity analysis or acknowledgment of alternate ABR return scenarios would strengthen the discussion (see Supplementary Material 1, Text S3, p. 25).
2. The classification of villages and prefectures by EOT probability is well visualized (Figure 7, p. 23). However, the summary of these findings in the Results section could be more concise. Consider adding a summary table mapping regions/prefectures to EOT probability categories, treatment history, and ATS recommendations. This can be provided as part of the supplementary materials.
3. The decision to rely on crude microfilarial prevalence (Methods, p. 7) is explained (due to missing standardized values), and a 0.99 correlation with standardized data supports this approach (Supplementary Figure S6, p. 24). Nonetheless, adding a brief note on how model fits were selected in villages lacking baseline data would improve transparency (also relevant to Text S7, p. 35–44).

Minor comments

- In the Abstract, consider revising “VC plus MDA substantially reduced prevalence” to “Combined VC and MDA interventions led to substantial reductions in prevalence...” to improve precision.
- When mentioning the Oti River and its tributaries, consider inserting a brief explanation of the significance of these areas as transmission foci. (Page 4)
- Specify that the assumption of ABR recovery post-VC is conservative, and note potential alternative scenarios briefly. (Page 8).
- Tables 1 and 2 are informative. Consider referencing Table S2 more explicitly for readers interested in prefecture-level intervention details.
- The model averaging approach over 100 iterations is described well; however, please clarify how model fits were determined in BMP-unknown villages.
- Figure 7 is visually effective. To enhance its operational utility, include a supplementary summary table (new Table S30?) consolidating ATS recommendations by prefecture.

- The use of "Special Intervention Zones (SIZ)" vs "SIZ" is not consistent. Recommend standardizing across text and figures.
- Consider simplifying terms such as "pseudo-equilibrium" (Discussion, p. 24) to improve accessibility for non-specialists.

(Remarks on code availability)

Reviewer #2

(Remarks to the Author)

The authors used the EPIONCHO-IBM tool to evaluate the end of transmission in Togo using ongoing intervention strategies. They noted that while vector control can support elimination, stopping it may lead to a resurgence in hypoendemic areas due to persistent infection in blackfly populations.

The study carefully evaluates site-specific scenarios and estimates elimination probabilities, classifying them as likely, unlikely, or very unlikely by 2030. This is a good demonstration of the utility of the EPIONCHO-IBM tool to support policy and decision-making.

Overall, I like the way the paper is structured and written, especially the presentation of results and the conclusions drawn. However, I find the paper too long and somewhat repetitive. The repeated reporting of similar results across sites makes it cumbersome to read. Some of these could be moved to the supplementary material, which I again believe is too crowded to follow along.

I have two suggestions:

1. Can the authors suggest policies based on their classification of areas with unlikely and very unlikely elimination probabilities? For example, how long might elimination take in those areas, and what alternative intervention strategies could be considered? A discussion along these lines would add value to the paper, especially in terms of how elimination could still be achieved in more challenging settings.
2. Can the authors include a brief discussion on how the EPIONCHO-IBM tool could be used to account for migration and deforestation? These factors may become increasingly important in end-game scenarios.

(Remarks on code availability)

I have cross checked the code and it works fine. It does have a README file.

Version 1:

Reviewer comments:

Reviewer #1

(Remarks to the Author)

I would like to thank the authors for their thorough and thoughtful responses to my comments on the original version of the manuscript.

The revised version is significantly strengthened. In particular, I appreciate:

- The detailed justification for the assumption on post-vector control biting rates, supported by regional entomological data;
- The addition of Table S31, which synthesizes programmatic recommendations by prefecture and strengthens the paper's relevance for policymakers;
- The clarification of how best-fitting endemicity scenarios were selected for villages without baseline data;
- The improvements in consistency and readability throughout the text, figures, and supplementary material.

These revisions have improved the clarity, transparency, and operational value of an already strong and timely manuscript. I have no further comments and fully support its publication.

Congratulations to the authors on this important contribution.

(Remarks on code availability)

Reviewer #2

(Remarks to the Author)

I thank the authors for addressing my concerns.

(Remarks on code availability)

Replies to Reviewers' Comments

REVIEWER 1

This manuscript presents a comprehensive evaluation of onchocerciasis elimination efforts in Togo through the application of the EPIONCHO-IBM transmission model. Using detailed village-level epidemiological data spanning nearly five decades, the authors simulate the impact of historical and current interventions across different endemicity settings and regions. The manuscript is a strong and policy-relevant contribution to the field of NTDs and offers critical insights into the heterogeneity of progress toward elimination of transmission (EOT) in Togo. The manuscript offers novel insights by combining long-term prevalence data with a validated individual-based model (EPIONCHO-IBM). It stands out for the breadth of the dataset (400 villages) and the policy utility of projecting EOT probabilities at sub-national level. The classification of regions into stop-MDA-ready and ATS-needing zones is especially useful for national programs and aligns well with WHO guidelines. Regarding model implementation, the inclusion of the EPIONCHO-IBM code link and open-access data enhances reproducibility. The authors also align their reporting with the PRIME-NTD principles (Supplementary Material 1, Text S6, p. 31), which is commendable.

Reply by Authors to general appraisal of the manuscript by Reviewer 1: We are very grateful for these appreciative comments and the value that the reviewer sees in our paper.

Comment 1.1 While the authors justifiably assume that annual biting rates (ABRs) return to pre-intervention levels post-vector control (post-VC) (Methods, p. 8), this assumption may underestimate the sustained impact of ecological or human-induced changes (e.g., deforestation, population movements, changes on human habits leading to decline in human-vector contact, etc.). A brief sensitivity analysis or acknowledgment of alternative ABR return scenarios would strengthen the discussion (see Supplementary Material 1, Text S3, p. 25).

Reply by Authors to 1.1: We thank the reviewer for highlighting this important issue. While our model assumes that annual biting rates (ABRs) fully recover to pre-intervention levels a year after the cessation of vector control (VC), we recognise that this as a conservative assumption which might not fully capture the impact of ecological or anthropogenic changes (e.g., deforestation, decreased biting rates, changes in human behaviour regarding exposure to vectors) over the years.

However, several entomological studies in Togo and other West African countries have documented rapid rebounds of ABRs to levels very similar to those recorded prior to vector control. For example, in the Mô River Basin of Togo, post-VC ABRs in 2015-2019 ranged from 12,000 to 16,000 bites/person/year, compared with pre-control ABRs of 16,000-47,000 (Supplementary Material 1, Table S4). Similarly, in areas of Burkina Faso bordering with Togo, ABRs rapidly returned to or surpassed pre-control levels within two years of stopping VC (e.g., Ziou Zabré: 30,739 post-control compared to 11,879 pre-control; Loaba: 8,617 post-control compared to 6,090 pre-control). Likewise, in Niger, the Goulbi River Basin experienced a resurgence to 14,000 bites/person/year in 1988, within one year of ceasing vector control in 1987, compared to a baseline value of 14,350 in 1978. Even in deforested areas such as the

Wawa River in Ghana, a water course shared with Plateaux Region in Togo, ABRs rebounded to pre-control levels after VC ended, reaching 2,400-4,000 bites/person/month compared to 2,800-4,100 bites/person/month prior to vector control.

Therefore, we believe that our assumption is supported by empirical evidence. Nevertheless, we acknowledge that this may not be the situation in all foci in Africa, and future modelling studies should account for secular trends in ABR if documented changes in transmission conditions warrant the modelling of these trends.

We have included text in our revised Discussion as follows:

“Although baseline ABRs were reduced during VC by its assumed efficacy, they were modelled as bouncing back to pre-control levels one year after VC cessation [31]. This conservative assumption is supported by several entomological studies conducted in Togo and other OCP countries. In the Mò River Basin, ABRs in 2015-2019 (12,000-16,000 bites/person/year) [35,41] were comparable to pre-VC values (16,000-47,000; Supplementary Material 1, Table S4). In areas of Burkina Faso bordering with Togo, ABRs returned to or surpassed baseline levels within two years of stopping VC (Loaba: 8,617 compared to 6,090; Ziou Zabré: 30,739 compared to 11,879) [7]. In Niger, the Goulbi River Basin experienced a bounce-back to 14,000 bites/person/year within one year after ceasing vector control, compared to baseline values of 14,350 [43]. Even in areas experiencing deforestation as in the Wawa River in Ghana (a water course shared with Plateaux), biting rates rebounded to pre-control levels after VC ended, reaching 2,400-4,000 compared to 2,800-4,050 bites/person/month [44]. Notwithstanding, deforestation, particularly in western Plateaux and southern Centrale [45], may have led to secular changes in vector density [46] not considered in the model. Future modelling studies should account for secular trends in ABRs when substantial changes in transmission conditions have been documented (see for instance [47]).”

Comment 1.2 The classification of villages and prefectures by EOT probability is well visualized (Figure 7, p. 23). However, the summary of these findings in the Results section could be more concise. Consider adding a summary table mapping regions/prefectures to EOT probability categories, treatment history, and ATS recommendations. This can be provided as part of the supplementary materials. “We also synthesised the results pertaining to “Elimination probabilities” ...

Reply by Authors to 1.2: We thank the referee for this invaluable suggestion. We have prepared an additional supplementary table (new Table S31) summarising, by prefecture, the likelihood category of reaching elimination of transmission (by 2024, 2027 and 2030), the current interventions and our recommendations. However, as modelling specific ATS would be beyond the scope of this paper, we base our recommendations on our previous modelling work and have included a footnote for Table 31 indicating some references.

We have included text in our revised Results as follows:

“Figure 7 illustrates the (categorical) likelihood of reaching EOT if ivermectin MDA stops in 2027. Table S31 summarises current control strategies and recommendations by prefecture.”

Comment 1.3 The decision to rely on crude microfilarial prevalence (Methods, p. 7) is explained (due to missing standardized values), and a 0.99 correlation with standardized data supports this approach (Supplementary Figure S6, p. 24). Nonetheless, adding a brief note on how model fits were selected in villages lacking baseline data would improve transparency (also relevant to Text S7, p. 35–44).

Reply by Authors to 1.3: We acknowledge the need for further transparency. We implemented the (median) Mean Squared Error (MSE), between observed and simulated microfilarial prevalence across model repeats, as a metric to identify the best-fitting endemicity and intervention scenarios (those with the smallest MSE values) for villages without baseline data. We have added the following to the Methods section:

“To identify the intervention scenarios followed by villages lacking BMP estimates, we used the (median) Mean Squared Error (MSE) between observed and simulated microfilarial prevalence trends across model repeats, with the smallest MSE values indicating the best-fit scenarios (Text S7, Table S7)”.

In Supplementary File 1 Text S7 (Modelled infection trends by region and Special Intervention Zone (SIZ) status for villages without recorded baseline microfilarial prevalence estimates of *Onchocerca volvulus* in Togo), we added the following (and included a new Table S7):

“The results presented here are based on trend analysis. Baseline microfilarial prevalence (BMP) estimates for these villages had not been recorded. The most likely baseline endemicity categories and intervention scenarios for these villages were inferred visually and by calculating the (median) mean squared error (MSE) between observed and simulated prevalence trends across 100 model repeats, with the best-fit scenarios indicated by the smallest MSE value.

For each region and endemicity level, the best-fit scenario for villages without BMP estimates was generally consistent with the scenarios identified among villages with recorded BMP estimates. Table S7 presents, for each region and endemicity category: (1) the scenario with the lowest MSE among villages with known BMP = ‘Best-fit for villages with BMP estimates’; (2) MSE values for villages without BMP when applying the same scenario as in (1) = ‘Best-fit for villages without BMP estimates’; (3) MSE values when the best-fit scenario for villages without BMP departed from (2) or there were no villages in (1) = ‘Alternative scenario for villages without BMP estimates’.

Comment 1.4 In the Abstract, consider revising “VC plus MDA substantially reduced prevalence” to “Combined VC and MDA interventions led to substantial reductions in prevalence...” to improve precision.

Reply by Authors to 1.4: Thank you for this suggestion. We had initially used the shorter version to remain within the recommended 250-word limit for the Abstract. We have now revised the sentence as suggested. The new version only slightly exceeds the word count, but we may have to revert to the original if we find a problem when copying the text of the Abstract into the corresponding box of the editorial system.

Comment 1.5 When mentioning the Oti River and its tributaries, consider inserting a brief explanation of the significance of these areas as transmission foci. (Page 4).

Reply by Authors to 1.5: In the main article, the main river basins associated with onchocerciasis transmission (Oti River Basin, including Kara, Kéran and Mô; and Mono River Basin) are mentioned and visualised in Figure 1; as well as mentioned in the Results section when presenting model output per region. We have, additionally, briefly explained the epidemiological significance of the main river courses contributing to onchocerciasis transmission in Supplementary Material 1, Text S1.2, as follows:

“Prior to control interventions, most of Togo was classified as meso- to holoendemic for onchocerciasis, with only the southernmost Maritime region classified as mostly hypoendemic (Figure S1B). The country’s two principal river basins, the Oti (also known as Pendjari or Oti/Pendjari, with main tributaries including the Kara, Kéran and Mô rivers) and the Mono (with tributaries such as the Amou, Anié, Kra, Haho, Ogou, Yoto and Zio rivers), had some of the highest annual biting rates in the country (reaching up to 100,000 bites/person/year, Table S4). Other, smaller river basins, in which onchocerciasis endemicity has been documented, include the White Volta (Volta Blanche) and the Todzie rivers, as well as tributaries of Lake Volta (Asukawkaw, Gban Hou and Kpaza Koue rivers) and Lake Togo (Haho, Yoto and Zio rivers).”

We have included the following in the Main Text:

“The blackfly-prolific Oti River Basin and its tributaries (Kara, Kéran and Mô) (Figure 1, Text S1.2), as well as at-risk hard-to-reach villages pose particular challenges to EOT [15,16].”

Comment 1.6 Specify that the assumption of ABR recovery post-VC is conservative, and note potential alternative scenarios briefly. (Page 8).

Reply by Authors to 1.6: We refer the reviewer to our **Reply by Authors to 1.1**.

Comment 1.7 Tables 1 and 2 are informative. Consider referencing Table S2 more explicitly for readers interested in prefecture-level intervention details.

Reply by Authors to 1.7: Table S2 is referred to in the Introduction and in the footnote of Table 1, as well as where relevant in the Methods and Results sections. Additionally, we have included the following in the Methods section:

“EPIONCHO-IBM was implemented across the four endemicity levels aforementioned within Togo’s five regions, considering their SIZ status and intervention history (Table 1). Table S2 provides intervention details at prefecture level.”

Comment 1.8 The model averaging approach over 100 iterations is described well; however, please clarify how model fits were determined in BMP-unknown villages.

Reply by Authors to 1.8: We refer the reviewer to our **Reply by Authors to 1.3**.

Comment 1.9 Figure 7 is visually effective. To enhance its operational utility, include a supplementary summary table (new Table S30?) consolidating ATS recommendations by prefecture.

Reply by Authors to 1.9: We refer the reviewer to our **Reply by Authors to 1.2**.

Comment 1.10 The use of “Special Intervention Zones (SIZ)” vs “SIZ” is not consistent. Recommend standardizing across text and figures.

Reply by Authors to 1.10: We appreciate this suggestion. In the revised manuscript, we have ensured consistency by defining “Special Intervention Zones (SIZ)” in full at first mention in the Abstract and Main Text, and using “SIZ” thereafter. In Tables, we use the full term (“Special Intervention Zones”) when first mentioned for clarity and to ensure the tables are self-explanatory.

Comment 1.11 Consider simplifying terms such as “pseudo-equilibrium” (Discussion, p. 24) to improve accessibility for non-specialists.

Reply by Authors to 1.11: We appreciate the reviewer’s concern about accessibility, but we feel that this term conveys well the dynamic behaviour of infection prevalence under conditions of intense transmission in our model “Pseudo-equilibrium” accurately describes the dynamic state under ongoing interventions in which the system oscillates around a pseudo-steady state which is lower than the endemic equilibrium but which indicates that prevalence will not decrease further under the prevailing transmission conditions. Fluctuations around the new, ‘pseudo-equilibrium’ result from repeated ivermectin treatment and the cyclical recovery of microfilarial prevalence when inter-treatment transmission is high under the assumptions of our model. For clarity, we have added a brief explanatory sentence in the Main Text:

“Conversely, modelled prevalence in areas with hyper- and holoendemic villages, such as the Oti River Basin (Savanes), and the Kéran and Mô River Basins (Kara and Centrale), declines to a pseudo-equilibrium (a seemingly steady state of infection maintained by the opposing effects of treatment and microfilarial prevalence recovery under conditions of intense inter-treatment transmission) since 2007 (Figures 2A-2B, 3C-3D, 4A)”

REVIEWER 2

The authors used the EPIONCHO-IBM tool to evaluate the end of transmission in Togo using ongoing intervention strategies. They noted that while vector control can support elimination, stopping it may lead to a resurgence in hypoendemic areas due to persistent infection in blackfly populations. The study carefully evaluates site-specific scenarios and estimates elimination probabilities, classifying them as likely, unlikely, or very unlikely by 2030. This is a good demonstration of the utility of the EPIONCHO-IBM tool to support policy and decision-making.

Reply by Authors to general appraisal of the manuscript by Reviewer 2: We thank the referee for this positive appraisal of our paper and the contribution that EPIONCHO-IBM can make towards policy and decision-making.

Comment 2.1 Overall, I like the way the paper is structured and written, especially the presentation of results and the conclusions drawn. However, I find the paper too long and somewhat repetitive. The repeated reporting of similar results across sites makes it cumbersome to read. Some of these could be moved to the supplementary material, which I again believe is too crowded to follow along.

Reply by Authors to 2.1: We recognise the manuscript is comprehensive due to the extensive data across regions and sites. We had already omitted or condensed some repetitive results and moved additional details to the Supplementary Material files. However, given the scope and complexity of the analysis, some repetition is necessary when presenting the results by region and Special Intervention Zone status to ensure clarity and completeness.

Comment 2.2 Can the authors suggest policies based on their classification of areas with unlikely and very unlikely elimination probabilities? For example, how long might elimination take in those areas, and what alternative intervention strategies could be considered? A discussion along these lines would add value to the paper, especially in terms of how elimination could still be achieved in more challenging settings.

Reply by Authors to 2.2: We thank the reviewer for this valuable suggestion, which echoes **Comment 1.2** by Reviewer 1. We have prepared a new Table S31 in Supplementary File 2, which summarises recommendations by prefecture based on our likelihood categories of elimination. Where elimination is unlikely or very unlikely by 2030, we recommend intensifying current interventions (e.g. switching from annual to biannual ivermectin treatment), or considering alternative treatment strategies (ATS). Our current analysis does not predict exact timeframes for elimination beyond 2030 in these settings, and further modelling would be needed to develop tailored ATS. However, we suggest some ATS (e.g. biannual moxidectin MDA) based on our previous modelling work, which we cite in the footnote of Table S31. Other ATS are also listed in the footnote. Table S31 is now referenced in the main text:

“Figure 7 illustrates the (categorical) likelihood of reaching EOT if ivermectin MDA stops in 2027. Table S31 summarises current control strategies and recommendations by prefecture.”

Comment 2.3 Can the authors include a brief discussion on how the EPIONCHO-IBM tool could be used to account for migration and deforestation? These factors may become increasingly important in end-game scenarios.

Reply by Authors to 2.3: We thank the reviewer for raising these important considerations. Regarding migration, we have recently used EPIONCHO-IBM to model the effects, on the introduction or re-introduction of infection, of an influx of immigrants with varying worm burdens arriving from an area with ongoing transmission into an onchocerciasis-free setting with local blackfly populations (new reference [48]). The effects of deforestation could be reflected in secular reductions of vector biting rates, or even vector disappearance in the case of simuliid species highly adapted to forest environments (such as *Simulium neavei* and *S. woodi*, in the *neavei* group) (described in reference [20]). Deforestation could also result in changes in vector distribution and species composition, favouring expansion of savannah members of the *damnosum* complex (reference [46]). We have added the following to the Discussion:

“Notwithstanding, deforestation, particularly in western Plateaux and southern Centrale [45], may have led to secular changes in vector density [46] not considered in the model. Future modelling studies should account for secular trends in ABRs when substantial changes in transmission conditions have been documented (see for instance [47]).”

“Also, EPIONCHO-IBM models closed populations, not accounting for movement of people and/or flies between villages or cross-border migration that could jeopardise EOT by re-introduction of infection from less-well controlled areas [42]. Recently, EPIONCHO-IBM has been used to model the effects, on introduction or re-introduction of infection, of an influx of immigrants with varying worm burdens arriving from an area with ongoing transmission into an onchocerciasis-free setting with local blackfly populations [48].”

Comment 2.4 I have cross checked the code and it works fine. It does have a README file.

Reply by Authors to 2.4: We thank the reviewer for confirming that the code provided works properly and is adequately documented.